# Fear of the unknown: Experience of frontline healthcare workers with coping strategies used to face the COVID 19 pandemic

**Gabriela Arango-Martinez** **, Laura Becerra Sarmiento** *, **Isabela Castaneda Forero, Laura Castaneda Carreno, Yazmin Cadena-Camargo**

Department of Preventive and Social Medicine, Pontificia Universidad Javeriana, Bogotá, Colombia

* lbecerra.ssa@gmail.com

## Abstract

The COVID 19 pandemic presented various challenges among health care workers, one of them being the impact it has on mental health. The psychological problems such as anxiety, depression, insomnia and stress, all consequences of the pandemic cause psychopathological outcomes reverberating negatively on the emotional well-being of health care workers. This study aimed to explore the experience of frontline healthcare workers (HCWs) during the COVID-19 pandemic in a middle-income country in Latin America and to identify the coping mechanisms they used to face stressful situations during this time. A qualitative study based on an interpretative paradigm was conducted allowing to examine complex, ambiguous and emotionally loaded topics to explore in detail the experience of frontline HCWs during the COVID-19 pandemic. Through convenience sampling eleven frontline HCWs were recruited to participate in semi-structured interviews. An inductive analysis was done with four pre-established categories: the experience of HCWs during the attention of COVID-19 patients, the experience during their own infection, the experience during the work reintegration and the coping mechanisms used. Our results show that fear and uncertainty predominated throughout the attention of COVID-19 patients. Participants used both coping strategies based on problem-solving efforts, such as routine changes, and emotional management efforts, like social support or psychological counselling. The choice of coping strategies was influenced by their personal beliefs, past emotional experiences, and prior medical formation. These findings provide public and private institutions insight for creating effective policies, based on the HCWs' preferences, to promote their psychological well-being.

## Introduction

The COVID-19 pandemic has exposed healthcare workers (HCWs) to stressful working conditions around the world [1]. Examples of these situations are the high demand of healthcare services, longer working hours, the allocation of scarce resources, shortage of personal protective equipment (PPE), an increased risk of infection, or the fear of transmitting the virus to family and friends [1–3]. These factors have had negative consequences on the mental health

**Data Availability Statement:** All relevant data are within the paper.

**Funding:** The authors received no specific funding for this work.

**Competing interests:** The authors have declared that no competing interests exist.

of frontline HCWs [4–6]. Studies conducted throughout the COVID– 19 pandemic have established that Chinese and Italian frontline HCWs are more likely to experience psychological problems such as anxiety, depression, insomnia, or stress [4–6]. Similarly, Colombian HCWs had higher prevalence of anxiety and depression than the general population [7].

The long-term consequences on the mental health of HCWs are yet to be determined. Nonetheless, studies have reported the presence of higher levels of stress, burnout, and depressive symptoms in frontline HCWs even after periods of time ranging from one to three years after the end of the 2003 SARS outbreak [8–10]. Thus, giving insights into the possible long-term consequences of the COVID-19 pandemic on frontline HCWs' mental health. Pandemics and epidemics can lead to psychopathological outcomes in HCWs [11]; therefore, it is important to establish how HCWs' mental health can be preserved during such times. A study in Austria identified that coping strategies, such as positive thinking and social support, had a beneficial impact on psychological life quality, well-being, perceived stress, depression, anxiety, and insomnia [12], highlighting the importance of adequate coping mechanisms to preserve mental health.

Lazarus and colleagues have proposed a cognitively oriented theory for stress and coping which suggests that psychological stress arises when the individual perceives a situation that exceeds his or her resources and endangers his or her well-being [13]. This model is based on two important phases: cognitive appraisal and coping [14]. Cognitive appraisal is subdivided into primary appraisal and secondary appraisal [14]. Primary appraisal is an evaluation of the potential harms and stressors, while secondary appraisal assesses the individual's ability and resources to cope with the situation [14]. After cognitive appraisal, coping takes place, which refers to the behavioural and cognitive efforts that are used to manage or reduce the demands generated by the stressful situation. Coping efforts can be aimed at problem management or emotional regulation [14]; moreover, their use gives rise to multiple outcomes of the coping process such as psychological well-being [15]

It has been determined that HCWs in high-income countries, like Japan or the United States, used coping mechanisms such as exercise, artistic activities, spiritual practices, or social support, during the pandemic [16,17]. Nonetheless, there is little information on the coping strategies used by HCWs in Latin America, a continent conformed by low- and middle-income countries. Therefore, the objective of this qualitative study is to explore the frontline HCWs' experience in Colombia, a middle-income country, during the COVID-19 pandemic to identify the coping strategies that were used by HCWs. The results of this article aim to help private and public institutions create effective strategies for maintaining HCWs' psychological well-being and preventing psychopathological outcomes during emergency situations, such as the COVID-19 pandemic.

## Methods

### Study design

This study used a qualitative design based on an interpretative study. This methodology was chosen as it is useful for examining topics that are complex, ambiguous, and emotionally loaded [18], allowing us to explore in detail the experience of frontline HCWs during the COVID-19 pandemic.

### Participants and setting

The present authors conducted the study during August 2021 in a teaching hospital in Bogotá, Colombia. The population included were frontline HCWs who had a positive test for COVID-19 during the pandemic. For the purpose of this study, frontline HCWs were defined as those

who directly treated patients with SARS- CoV 2 infection. Moreover, the motive of choosing HCWs with a positive COVID-19 test was to document the experience during their infection with COVID-19 and the process of work reintegration. All participants were of legal age, above 18 years, by the Colombian law.

## Data collection and analysis

The research group was constituted by members of the School of Medicine of the Pontificia Universidad Javeriana. Participants were invited to be part of the study through convenience sampling until reaching the saturation point. We recruited eleven participants with various roles within the hospital such as emergency doctors (ED), paediatricians, registered nurses, nurses' assistants, respiratory therapists, and spiritual companions. There were four male and seven female participants. Data was collected through semi-structured interviews permitting to explore the HCWs' experience during the treatment of patients with COVID-19, their own infection, their work reintegration, and inquire about the coping efforts and mechanisms used during the pandemic. The interviews took place in the hospital facilities and lasted between twenty minutes to an hour. They were conducted in Spanish and transcribed verbatim, however, for the purpose of the article, quotes were translated into English by researchers. An inductive analysis was done with four pre-established categories: the experience of HCWs during the attention of COVID-19 patients, the experience during their own infection, the experience during the work reintegration process and lastly the coping mechanisms used.

## Ethical framing

Research and ethical approval were granted by the Ethics Committee of the School of Medicine of the Pontificia Universidad Javeriana, Bogotá (FM-CIE-03359-20, Act 09/2020). Moreover, the research protocol and methods were consistent with Colombian Law. Prior to the interviews, participants were asked to sign an informed consent and were fully informed about all aspects of the project including its objectives and methodology. They were notified that they could withdraw at any moment without providing reasons, and that eventual withdrawal would not affect their jobs in any way. Anonymity and confidentiality were ensured by removing the participants names and coding the interviews with numbers.

## Results and analysis

The results portray the experiences of eleven frontline HCWs during the COVID-19 pandemic. The analysis was based on four pre-established dimensions: 1) the experience, emotions, and implications of treating patients with COVID-19, 2) the HCWs experience during their own infection with Sars- CoV 2; 3) the experience during work reintegration after their recovery and 4) the coping mechanisms used.

## (1) HCWs Experience, emotions, and implications of treating patients with COVID-19

The first tangible change that participants identified due to the pandemic were the biosecurity protocols that were established for the treatment of COVID-19 patients. Participants shared a common experience as they all had to learn how to properly use the personal protection elements (PPEs) and correctly follow biosecurity and isolation protocols, such as handwashing and uniform change. These strategies helped participants to feel safe during the pandemic. However, at the beginning these mechanisms limited interactions with patients which altered the dynamics of the doctor-patient relationship. An ED portrays this experience as follows:

*"Fue muy duro el cambio de usar tapabocas permanentemente, porque uno tiene que ser sincero y la verdad antes de la pandemia uno no era muy disciplinado con esto del tapabocas [. . .] Con el Covid esto fue el inicio del uso del N-95, compra de máscaras, usar escafandras, usar batas y usar múltiples implementos qué no estábamos acostumbrados que en muchas ocasiones nos afectaba escuchar bien y nos faltaba sobre todo hablar con el paciente."*

- ED (male).

[*"The permanent use of the face mask was initially very difficult, I must be honest, and the truth is that before the pandemic we were not very disciplined with its use. [. . .] With Covid we started to use the N-95 face mask, body suits, disposable medical gowns, and more elements that we were not used to using, many times they* (PPEs) *did not allow us to hear properly, we missed talking to the patients; that was the most difficult thing."*]

- ED (male).

Participants appraised the institution's proper adequation of PPEs for their workers and the rapid creation of improvised intensive care units (ICU) to accommodate critically ill patients. Nonetheless, the increase in critical ill patients augmented the workload of HCWs and exposed the unpreparedness of the healthcare system to face the pandemic. During the interviews, participants, especially doctors, frequently mentioned the scarcity of material and human resources during the outbreaks. The duality between wanting to help and realizing there were not sufficient resources to treat patients, had a negative impact on their mental health, mainly by increasing stress levels amongst HCWs. The experience of an ED exemplifies this common sentiment:

*"Mi estrés, desde el punto de vista mental en el ejercicio médico se presentó, fue ya cuando arrancamos el pico en forma tanto el primero como el Segundo, que llega un punto que uno decía: 'Dios mío va a llegar otro paciente y ya se nos acabaron los recursos, ya no hay ventiladores, ya no hay camas en la UCI, ya no hay donde conectar más pacientes' Entonces, ese temor de pensar que no lleguen más pacientes por favor, que al próximo que llegue ¿qué le vamos a ofrecer? Era muy estresante"*

–ED (female).

[*"Mental stress appeared at the peak of the first and second outbreak. There was a point where I would say: 'Oh God, another patient is coming, and we are out of resources, there aren't any ventilators or beds, there is no place to connect another patient.' So, there was fear, and I thought please let no more patients come because what are we going to offer the next one? It was very stressful"*]

–ED (female).

As highly trained HCWs, participants were used to guide treatments based on the best available evidence and the highest standards of care. Nonetheless, the SARS-Cov 2 virus was an unknown pathogen that generated a disease without precedence or knowledge regarding transmission nor treatment. Therefore, participants had to learn and adapt quickly to new treatment protocols and procedure techniques whilst dealing with scarce resources. The lack of information within the general population led to mistrust and doubts of treatments and medical decisions. Such mistrust created a stressful environment in which HCWs, especially doctors, were constantly questioned about their medical conducts. Due to the general

population's scepticism, participants experienced discomfort and anger, because while they were trying to treat critically ill patients with the best available evidence, they were facing constant backlash. An ED experience depicts the participants struggles:

*"Cuando llegaban tan graves, la explicación apenas llegaban uno le dice a los familiares: 'Toca intubar' y la manifestación de la gente era: '¿Cómo así? ¡Para ustedes ahora todo es intubar!'. Entonces era muy estresante, o sea más que sentirse uno agredido, era muy estresante, muy molesto que la gente no comprendiera que precisamente eso era lo clásico del virus, que la gente aguanta hasta que ya se está muriendo, por eso cuando la gente ya está muriendo es que llega y por eso es que toca incubarlos [. . .] Pues darle todo esa explicación a la gente fue muy desgastante hasta un punto en que uno ya decía ya no les voy a explicar más nada porque es demasiado desgastante y, es ofensivo que uno se tomó todo el tiempo para explicarle a la gente, uno está super cansado, les estás ofreciendo un recurso que a muchos otros tocaba decirles que no había, y al que le estamos diciendo que si hay, pues tenga ese rechazo"*

–ED (female).

[*"When patients arrived in a critical condition, the initial explanation, just as they arrived, to their relatives was: 'We must intubate' and their reaction was: 'What? Now everything is about intubating!' It was incredibly stressful, more than feeling attacked, it was stressful, and very annoying that people didn't understand that was the virus' classical behaviour, that people would wait to the last minute until they were dying to come and that's why we had to intubate [. . .] Having to explain why certain medical procedures were necessary was exhausting to the point I didn´t try to explain. It was very exhausting and very offensive; I was taking the time to explain, I was very tired, I was offering them a scarce resource that I had to deny to many others because there weren't enough, and they still rejected it"*]

–ED (female).

Many participants faced discrimination and stigmatization outside their working institution. Rejection acts were driven primarily by the general population's fear of infection. This shows that the impact of the ignorance surrounding the virus transcended the working environments of HCWs. The effect of these acts was different for each participant. Some experienced hurt and anger for being mistreated while others understood the fear of the general population and were not affected by it. For the participants who did not experience stigmatization or rejection acts, they faced constant fear of them. The experience of a nurse assistant and a paediatrician exemplifies these feelings:

*"El medio de transporte era el Transmilenio. La gente cuando subía era algo imprudente y lo primero que hacían era desplazarse a las últimas sillas o a las primeras sillas, menos estar al lado de nosotros [. . .] Eso dolía bastante porque era vulnerar de pronto la integridad de la persona."*

–Nurse assistant (female).

[*"My means of transportation was the Transmilenio (*public bus*). The people that got in were imprudent, the first thing they did was move to the last or first chairs, anywhere but close to us [. . .] This used to hurt a lot because it was like violating our integrity"*]

–Nurse assistant (female).

*"En mi edificio no tuve ningún inconveniente, pero me pasaba que yo misma sentía como miedo y decía como mejor no me voy en uniforme hacer mercado, llego a mi casa me baño y salgo para que no sepan que soy médica, o si alguien se sube al ascensor en mi edificio va a querer salirse y no irse en ese viaje conmigo."*

–Paediatrician (female).

[*"I didn't have any inconvenience in my building. What happened to me was that I felt fear to go in uniform to the supermarket for example. I wanted to get home take a bath and change so they didn't know I 'm a doctor. Or I would also think that if someone gets in the elevator with me, that they will prefer to get out rather than to take this ride with me"*]

–Paediatrician (female).

Participants experienced similar feelings during the attention of Covid-19 patients, with fear and uncertainty predominating throughout the interviews. These feelings were generated by the risk of being infected while working, not knowing how the virus could affect them, but mostly the risk of infecting their families and friends. Moreover, sadness and impotence were also a common feeling for participants as they had to face the high death rates and the solitude of the dying patients. About this an ED and a registered nurse shared:

*"Un mar de incertidumbres, dudas, preocupaciones no sobre la salud de uno, sino de los familiares que estaban alrededor de uno."*

–ED (male).

[*"I experienced a lot of uncertainty, doubts and worries, not about my own health, but the health of relatives or people around me"*]

–ED (male).

*"Eso de verdad es muy triste que usted se muera y no pueda ver a su familiar ni a nadie, darle una sepultura digna""*

–Registered nurse (female).

*"It is really sad that when someone dies, their family and friends cannot see them, they cannot give them a proper burial"*

–Registered nurse (female).

Nevertheless, there were participants who did not experience an increase on their stress levels due to their work specialty background. Prior to the pandemic they were used to confront stressful situations such as the ones in the emergency room, which allowed them to manage stress while attending COVID-19 patients. An ED portrays his experience as follows:

*"Realmente el grupo de emergenciólogos, debo decir, todo el mundo estuvo muy tranquilo, porque en nuestra actividad nosotros tenemos que lidiar con muchas cosas, con la muerte, con situaciones muy difíciles para los pacientes, entonces básicamente una barrera y un autocontrol muy alto comparado con otra especialidad."*

–ED (male).

*["I have to say that the ED team remained very calm; because during our daily activities we must deal with many things such as death or difficult situations for patients, we basically had a very good auto control during the pandemic compared to other specialties."]*

–ED (male).

Overall, participants were exposed to the same conditions while attending COVID-19 patients. Although every experience was unique, several fears and emotions where shared between the different frontline HCWs in our study. Treating and accompanying COVID-19 patients is a different experience from being directly infected by the virus. Unfortunately all of our participants were directly infected by the SARS-COV2 virus. The next dimension will explore their personal experience of being infected.

## (2) Experience during their own infection

All participants were infected during working hours and had a clear source of contagion. Participants had seen the conditions in which patients were treated and knew the uncertainty of the course of the disease. Therefore, their fears during their own infection were based on the experience of treating COVID-19 patients. Most participants expressed fear of needing an ICU, of being hospitalized without their families' companionship and of dying alone. A paediatrician narrated her experience as follows:

*"Me dio mucho miedo, muchísimo miedo, solo el hecho de ir a la clínica a urgencias y que mi esposo me llevara y no lo dejaran entrar, era una despedida de esas que uno no sabe si uno volvía a salir o si no y que uno se iba a ir solo"*

–Paediatrician (female).

*["I was afraid, very afraid of going to the hospital, and of them not letting my husband in with me because I was saying goodbye without knowing if I was coming out or not."]*

–Paediatrician (female).

Although similar fears were experienced by participants during the infection, during the interviews it was clear that being infected had a bigger impact on emotional and mental health for those participants who lived with family and relatives. They feared the possibility of infecting their families and dealing with the guilt of exposing them to the virus. Moreover, for participants who are parents, guilt was augmented as they could not fulfil their parental duties due to the physical symptoms they were experiencing. The latter was evident in a nurse assistant testimony:

*"Fue terrible porque en esos últimos días yo no pude alimentar a mi bebé, ni siquiera para poderlo alzar. Entonces todo eso, es cuando uno empieza a echarse reproches[. . .]. Al ver yo todo eso me sentía culpable de saber que todo eso estaba pasando por uno"*

–Nurse assistant (female).

*["It was terrible because I couldn't feed my baby, not even hold him. [. . .] Seeing all this and knowing it was happening because of me made me feel guilty "]*

–Nurse assistant (female).

After the infection participants had a full recovery, none experienced any physical complications allowing them to return to their jobs as soon as the quarantine was over. However, the work reintegration experience was different for each participant.

### (3) Experience of work reintegration

Participants had multiple perspectives towards work reintegration. For some, reincorporation was characterized by an increase fear of reinfection, which led to the constant evaluation of measures for preventing the spread of the disease. These participants became more aware of possible contagion sources and were stricter adhering to safety protocols. For one participant the fear of being reinfected and the possibility of reinfecting her family led to job resignation. On the other hand, some participants were calm and felt safer during work reintegration. The difference between perspectives was mainly because the latter evaluated their infection in a positive way, they focused on the fact that there were no complications, and their fears did not become real. Thus, allowing them to carry out their duties coming from a place of safety and calmness. The following testimonies portray the opposite perspectives while reintegrating the working environment:

*"¡Yo me retiré! Yo me retiré porque tenía mucho miedo por lo mismo que me informaban de la reinfección y yo decía yo ya me recuperé una vez, la primera vez, no sabemos cómo mi familia tome la segunda, el segundo contagio."*

–Assistant nurse (female).

[*"I resigned*! *I resigned because I feared what I was hearing about the reinfection, I had already overcome the infection once, nobody could predict how me, or family would respond to a second infection".*]

–Assistant nurse (female).

*"Creo que eso también me ayudó a llegar fortalecida y creo que era obvio porque yo llegué más tranquila como bueno ya me dio y ya no me agrave entonces vengo atender al paciente con más tranquilidad y eso si lo noté."*

–Paediatrician (female).

[*"I started work more relaxed because I thought I already had it and I didn't have any complications, so I went to see the patient more relaxed, with less fear"*]

–Paediatrician (female).

No matter the experience on work reintegration, most participants agreed that being a patient changed their patient–doctor relationship with the COVID-19 patients. Participants highlighted that they were able to understand the fears and worries of their patients, which increased empathy levels and sensibility. This led to treating patients in a more humane way. A registered nurse and an assistant nurse commented as follows:

*"Yo creo que cuando uno vive la experiencia [. . .] se vuelve más sensible a todo eso, entonces yo creo que sí nos cambia el chip que tenemos [. . .] creo que es volverse más humano"*

–Nurse assistant (female).

["*I think that when one goes through the experience [. . .] one starts being more sensible to everything, [. . .] I think it did change my mindset, [. . .] I became more human*"]

–Nurse assistant (female).

"*Yo creo que eso me sirvió, y en estar más tranquila y darles ese acompañamiento y apoyo*"

–Registered nurse (female).

["*I think it was useful for me to start giving them companionship and support they needed*"]

–Registered nurse (female).

As seen above participants had to deal with many stressors while attending patients with COVID-19, during their infection and work reintegration, which took a toll on their mental health. Therefore, to maintain mental health participants used a variety of tools.

## (4) Coping strategies

Through the interviews it was evident the importance of support from family, friends, and co-workers to the point that social support became the most common coping strategy during the attention of COVID-19 patients and during the infection period.

"*Pero sí, yo pienso que durante todo el proceso, la familia, el hospital, el entorno y el grupo de trabajo ha sido muy importante*"

–Respiratory therapist (female).

["*Family, the hospital and my colleagues have been really important through this process*"]

–Respiratory therapist (female).

Generally, social networks allowed participants to share feelings, be heard and to be accompanied through the process. Through shared experiences work colleagues provided companionship and understanding, which reduced distress and loneliness while working. Moreover, they helped create a safer working environment by helping protect each other from infection. Family and friends were vital for support and solidarity which, in many cases, allowed participants to be at peace with their jobs. Two testimonies from a nurse assistant and a respiratory therapist portray the importance of these roles:

"*Yo creo que mi red de apoyo fue mi familia porque nunca me criticaron el hecho de estar trabajando en UCI donde teníamos 100% COVID toda la unidad.*"

- Nurse assistant (female).

["*My support was my family, they never critiqued me for working in the ICU, even when we had 100% occupation of COVID-19 patients.*"]

–Nurse assistant (female).

"*A veces en un afán, en un código uno entrar y decirle al doctor que le falta la careta. Ese apoyo como grupo asistencial ha sido muy grande*"

Respiratory therapist (female).

[*"Sometimes, when we were in a rush in a cardiac arrest event, I would tell the doctor that he was missing his face shield'. That kind of support as a team has been huge"*]

–Respiratory therapist (female).

As the main source of stress for participants were the conditions surrounding the attention of COVID-19 patients, different coping strategies were created institutionally. Participants mentioned the creation of Balint groups, the availability of individual psychological counselling and mindfulness sessions. It was evident that participants appreciated the creation of these tools as it was a manifestation of concern and support for their mental health. Although, there was no pattern in the use of coping mechanisms, females, especially nurses, relied on psychological counselling more than males. A female paediatrician manifests the following:

*"El hospital y la universidad todo el tiempo nos ofrecían ayuda, siempre sabíamos que tenías dónde consultar en caso de que necesitáramos y además de eso se programaban y todavía se programan sesiones de mindfulness o de relajación para ayudar a la gente"*

–Paediatrician (female).

[*"The hospital and the university were continuously offering help, we always knew where to go when we needed help and they scheduled, and still do, mindfulness and relaxation sessions."*]

–Paediatrician (female).

Individually participants also used different coping mechanisms which were dictated by their likes and beliefs, but all with the purpose of improving their mental health. Among these, participants referred using exercise, physical activity, hobbies such as singing, and spirituality through praying or meditation. Moreover, as participants identified certain sources of stress, they made routine adjustments to avoid or eliminate them. For example, some stopped using public transportation to avoid discriminatory acts, others changed their living conditions to lower the risk of transmitting the virus to their families. The following testimonies exemplify the use of these tools and routine adjustments:

*"Yo siento que hacer ejercicio, orar y rezar eso fue lo que me ayudó también."*

–Registered nurse (female).

*"I feel that doing exercise and praying is what helped me the most"*

–Registered nurse (female).

*"Me quedé viviendo solo siete meses, porque mi esposa y mis dos hijos los mande a una finca con todas la comodidades"*

–ED (male).

[*"I lived alone for seven months, I sent my wife and children to a country house with all the commodities."*]

–ED (male).

## Discussion

The results of this qualitative study suggest the pandemic exposed HCWs to high stress situations, that generated the development and use of strategies to cope with stress. According to our knowledge, there are few studies that explore the coping strategies used by healthcare workers during the pandemic in Latin America.

The following contributing factors for stress and decreased psychological well-being were identified: changes in the working environment, the health condition of patients, the fear of infection and of infecting others, stigmatization, the fear of the complications of the COVID-19 disease and of relapse, and self-blame for exposing family members to the virus. These results agree with existing literature. Work conditions reported by participants, such as the longer working hours and the lack of human resources, have been associated with a higher risk of developing stress and anxiety in Indian and Chinese HCWs [19–21]. Similarly, the fear of infection and of infecting others, the uncertainty generated by the patients' health condition, and the impotence regarding the high mortality rates, have been recognized as contributing factors to stress in Chinese HCWs [22]. Stigmatization and discrimination are common acts towards HCWs during pandemics and epidemics [23–26] and have been associated in Colombian HCWs and Korean workers with anxiety and depression [24,27] Moreover, the fear of complications during infection and the fear of relapse when returning to work were common feelings among recovered HCWs [28–30].

ED participants reported lower levels of stress while treating COVID-19 patients, proving that participants were not equally affected by the pandemic. Folkman (1984) proposes that primary appraisal, can be influenced by the familiarity of the event [14]. Therefore, as ED are constantly exposed to stressful situations in the emergency rooms, their initial evaluation of the pandemic could have not been regarded as too endangering to their personal well-being as to other participants. Moreover, constant exposure to stressful situations could have propitiated ED to develop coping strategies that helped maintain psychological well-being during the pandemic. These results differ from literature as it has been reported that ED had higher stress levels during the COVID-19 pandemic [31]. However, comparisons were made between frontline ED and HCWs who did not treat COVID-19 patients, thus possibly explaining the discordance in results.

Our results show that participants were able to identify and use different coping mechanisms to face stressful situations during the pandemic. Although the cognitively theory of stress and coping divides coping forms into emotion management and problem-solving efforts, they are not mutually exclusive and are usually used jointly when facing stressful situations [14]. Nonetheless, individuals usually preponderate the use of problem-solving efforts when situations are appraised as changeable, and emotion management when they are appraised as non-changeable [14]. We identified that problem-solving-oriented efforts were used by participants to face the fear of infecting relatives and the stress generated by discriminatory acts. To cope with stress, participants made lifestyle and routine changes, such as changing their means of transportation, using civilian clothes when leaving the workplace or living separately from their families. Similar strategies have been reported in a qualitative study conducted in English HCWs [32]. Moreover, knowing that one's family is safe has been identified as a protective factor against stress among Chinese HCWs [22].

The emotion-management-orientated coping strategies used by the participants of this study were social support, institutional tools, and personal tools. In accordance with our results, different studies have identified social support as a common coping strategy among HCWs during the pandemic [22,33–35]. During the first outbreak, support from family and colleagues was identified as a motivational factor for HCWs for continuing their jobs [36]. Zou

et al. (2022) determined that social support moderated the indirect effects of occupational stressors on anxiety and depression on frontline HCWs during the pandemic [37]. They explain their results through Cohen's stress-buffering model of social support that argues that social support might reduce the severity with which stressors are perceived and reduce the impact of stress appraisal in the individual [38].

Institutional tools orientated towards emotion management such as Balint groups, mindfulness meditation sessions and psychological support were also used by participants. Balint groups had beneficial consequences on HCWs' well-being during the pandemic as they increased perceived social support and connectivity [39], increased resilience scores, and reduced scores on the Corona Disease Anxiety Scale [40]. In their study Shechter et al. (2020) identified meditation as the fifth most common coping strategy in HCWs [16]. This tool can be useful as it may improve anxiety and depression [41]. Moreover, the implementation of mindfulness programs and meditation for HCWs has been associated with a better perception of quality of sleep and a reduced sense of loneliness [42], as well as the reduction of burnout syndrome, specifically the category of emotional exhaustion [43]. The importance of psychological counselling for Chinese HCWs differed depending on their role: most nurse participants rated it as a *very important/important* coping mechanism while medical doctors considered it as *important / slightly important* [22]. These findings are similar to ours as participants who referred individual psychological counselling as a coping mechanism were mostly nurses.

In this study, the use of practices such as exercise, prayer, and singing, was very rare. These results contrast with literature, as studies conducted in Palestine and the United States identified that exercise and prayer were the preferred strategies for the management of stress by HCWs during the pandemic [16,44]. Big cities in Colombia, like Bogotá register the lowest percentages of religious practitioners and the highest percentage of agnostic and atheists [45], statistics that could justify why religion was not used as a main coping strategy. Also, long working hours can interfere with the practice of exercise by HCWs [32], which could explain our results as the pandemic increased working hours for participants.

## Limitations

The group of researchers is part of the participants' hospital community which could have limited their responses to certain questions. To minimize the impact, rapport, and the creation of a confidential atmosphere was used. Participants were constantly assured that their responses were not going to affect their job position or conditions.

## Conclusion

The interpretative approach of this study allowed the identification of coping mechanisms used to face stressful situations during the pandemic by HCWs of a middle-income country in Latin America. In general, HCWs used a wide variety of coping strategies to confront the stressful and uncertain environment they were exposed to. We identified that problem-solving and emotional management efforts were used in conjunction by HCWs. Not only did their personal beliefs and past emotional strategies played a role in selecting the coping mechanisms, but their prior medical formation also played a key role on how they confronted the COVID-19 outbreak. Even though some participants shared the same medical specialty, every personal experience narrated throughout the interviews was completely unique in both the emotional aspect and impact on mental health. Nonetheless, fear was one of the most collective emotions felt by frontline HCWs during this time as they faced an undiscovered disease course and high risk of contagion in which each day brought a new teaching. Participants learned from

personal and external experience about the extent and management of the disease. The coping strategies used by HCWs were put to the test throughout the pandemic and were exemplified in the process of work reintegration as they had the personal experience of facing both sides of the disease: one being the patient and the other the HCW. The opportunity to interview frontline HCW using a qualitative study allowed to examine complex, ambiguous and emotionally loaded topics to be able to identify the coping strategies that were used by HCWs. These results provide public and private institutions insight for creating effective policies, based on the HCWs' preferences, that promote their psychological well-being.

## Acknowledgments

We would like to thank Dr. Luis Fernando Gomez (Pontificia Universidad Javeriana) for his valuable comments and support.

## Author Contributions

**Conceptualization:** Gabriela Arango-Martinez, Laura Becerra Sarmiento, Isabela Castaneda Forero, Laura Castaneda Carreno, Yazmin Cadena-Camargo.

**Data curation:** Gabriela Arango-Martinez, Isabela Castaneda Forero.

**Formal analysis:** Gabriela Arango-Martinez, Laura Becerra Sarmiento, Isabela Castaneda Forero, Laura Castaneda Carreno, Yazmin Cadena-Camargo.

**Funding acquisition:** Yazmin Cadena-Camargo.

**Investigation:** Gabriela Arango-Martinez, Laura Becerra Sarmiento, Isabela Castaneda Forero, Laura Castaneda Carreno.

**Methodology:** Gabriela Arango-Martinez, Laura Becerra Sarmiento, Isabela Castaneda Forero, Laura Castaneda Carreno, Yazmin Cadena-Camargo.

**Project administration:** Gabriela Arango-Martinez, Yazmin Cadena-Camargo.

**Resources:** Yazmin Cadena-Camargo.

**Software:** Yazmin Cadena-Camargo.

**Supervision:** Gabriela Arango-Martinez, Laura Becerra Sarmiento, Yazmin Cadena-Camargo.

**Validation:** Gabriela Arango-Martinez, Laura Becerra Sarmiento, Yazmin Cadena-Camargo.

**Visualization:** Gabriela Arango-Martinez, Laura Becerra Sarmiento.

**Writing – original draft:** Gabriela Arango-Martinez, Laura Becerra Sarmiento, Isabela Castaneda Forero, Laura Castaneda Carreno.

**Writing – review & editing:** Gabriela Arango-Martinez, Laura Becerra Sarmiento, Yazmin Cadena-Camargo.

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
