## [Decision Letter · Decision Letter 0]

10 Apr 2024

PGPH-D-23-02256

Fear of the unknown: Experience of frontline healthcare workers with coping strategies used to face the COVID 19 pandemic.

Dear Dr. Becerra Sarmiento,

Thank you for submitting your manuscript to PLOS Global Public Health. After careful consideration, we feel that it has merit but does not fully meet PLOS Global Public Health’s publication criteria as it currently stands. Therefore, we invite you to submit a revised version of the manuscript that addresses the points raised during the review process.

This is an important qualitative study exploring in detail the experience of frontline HCWs  of a middle-income country during the COVID-19 pandemic.

Interesting to note that the HCWs learned from personal and other experience to face the difficult situation.

The impact from this will help in the preparedness for the next pandemic and also to evolve strategies

We look forward to receiving your revised manuscript.

Kind regards,

Suma Krishnasastry, MBBS, MD,DNB

Academic Editor

Journal Requirements:

2. We have noticed that you have uploaded Supporting Information files, but you have not included a list of legends. Please add a full list of legends for your Supporting Information files after the references list.

Additional Editor Comments (if provided):

Reviewers' comments:

Reviewer's Responses to Questions

**Comments to the Author**

1. Does this manuscript meet PLOS Global Public Health’s publication criteria? Is the manuscript technically sound, and do the data support the conclusions? The manuscript must describe methodologically and ethically rigorous research with conclusions that are appropriately drawn based on the data presented.

Reviewer #1: Yes

Reviewer #2: Partly

Reviewer #3: Yes

2. Has the statistical analysis been performed appropriately and rigorously?

Reviewer #1: Yes

Reviewer #2: N/A

Reviewer #3: N/A

3. Have the authors made all data underlying the findings in their manuscript fully available (please refer to the Data Availability Statement at the start of the manuscript PDF file)?

Reviewer #1: Yes

Reviewer #2: Yes

Reviewer #3: No

4. Is the manuscript presented in an intelligible fashion and written in standard English?

Reviewer #1: Yes

Reviewer #2: Yes

Reviewer #3: No

5. Review Comments to the Author

Reviewer #1: The journal meet the PLOS Global Public Health’s publication criteria. Although there are no quantitiative data to review, the methodology and discussions were properly carried out in line with acceptable standards for a qualitiative research. The authors used appropriate research language; with clear statements and proper writing styles. The in-line referencing were correctly done and reflects rigorous literature review. The article makes for easy reading. The conclusion aligns properly with the methodlogy.

Reviewer #2: I read this article with interest.

However, the sampling and collection method suffers from a few shortcomings that could be a source of bias (choice of site, criteria for including participants). the choice of study site (health centre, hospital) and the criteria for recruiting participants should be included and detailed

Reviewer #3: Thank you for an insightful article.

A few observations were made.

1. There was no introduction in the abstract. It began with the aims of the study.

2. The are some grammatical errors which will need to be worked on. The following are some of the phrases I think need to be worked on:

"... the experience of HCWs during the attention of COVID-19 patients...."

"... conducted the study during August 2021....."

"..... at the beginning they imposed a challenge....."

"Such rejection acts were primary due to the fear..."

"More sadness and impotence were also a common feeling...."

"It is clear that to attend patients is very different...."

".... return to their jobsassoon a quarantine. "

" ...being more sensible to everything. "

6. PLOS authors have the option to publish the peer review history of their article (what does this mean?). If published, this will include your full peer review and any attached files.

**Do you want your identity to be public for this peer review?** For information about this choice, including consent withdrawal, please see our Privacy Policy.

Reviewer #1: **Yes: **Prince Obinna Anyanwu

Reviewer #2: No

Reviewer #3: No

---

## [Editor Report · Decision Letter 1]

2 Jul 2024

Fear of the unknown: Experience of frontline healthcare workers with coping strategies used to face the COVID 19 pandemic.

PGPH-D-23-02256R1

Dear Miss Becerra Sarmiento,

We are pleased to inform you that your manuscript 'Fear of the unknown: Experience of frontline healthcare workers with coping strategies used to face the COVID 19 pandemic.' has been provisionally accepted for publication in PLOS Global Public Health.

Best regards,

Suma Krishnasastry, MBBS, MD,DNB, FRCP (Edin)

Academic Editor